# Improving the Efficiency of Downhole Uranium Production Using Oxygen as an Oxidizer

Bayan Rakishev [1], Zhiger Kenzhetaev [1], Muhametkaly Mataev [2] and Kuanysh Togizov [1,*]

1   K.I. Satbayev Kazakh National Research University, 050013 Almaty, Kazakhstan
2   Institute of High Technologies LLP, NAC Kazatomprom JSC, 050013 Almaty, Kazakhstan
*   Correspondence: k.togizov@satbayev.university; Tel.: +7-7021710995

**Abstract:** The features occurring during borehole uranium mining in deposits with low filtration characteristics, as well as the conditions and reasons for the reduction of geotechnological parameters of uranium mining by the well are considered in this study. Core material samples were taken from the productive horizon of the Chu-Sarysui province deposit and granulometric compositions were established. The contents of uranium, aluminum, calcium, magnesium, iron and carbonate minerals in the samples were determined by atomic emission spectroscopy. The X-ray phase analysis method established the features and quantitative and qualitative characteristics of ore-containing minerals. A special technique has been developed for conducting experiments in laboratory conditions using core samples, where the intensity of uranium leaching in tubes is determined. The results of laboratory studies are analyzed and discussed and graphs are constructed, to show the dependencies of change in: the filtration coefficients of $K_f$; the uranium content in solution; the extraction coefficient; and the specific consumption of sulfuric acid on the values of L:S (the ratio of liquid to solid) in the experiments. The effectiveness of using a mild acidity regime, with the addition of oxygen as an oxidizer, is determined and shown. The values of the uranium content in the productive solution, with the addition of oxygen as an oxidizer, reached 220 mg/L, which exceeds the design parameters. The results of uranium extraction from ore show a positive trend, reaching 68%, with L:S from 1.7 to 3.0, low acidity values and the addition of oxygen as an oxidizer. The specific consumption of sulfuric acid reaches the minimum values when using leaching solutions with reduced acidity of 26 kg/kg. The obtained results, on the flow rate of the solution in the tube, the extraction of uranium from ore and the specific consumption of sulfuric acid, indicate a decrease in sedimentation in a porous medium and increased filtration characteristics, with reduced acidity values in the leaching solution.

**Keywords:** downhole uranium mining; X-ray phase studies; particle size characteristics; uranium leaching; tube

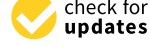



## 1. Introduction

Borehole mining of minerals, in particular uranium, involves the dissolution of a useful component by a moving solvent stream at the location of the ore body, followed by the removal and lifting of the formed compounds to the surface [1–3]. The positive aspects of the use of sulfuric acid solutions at enterprises in Kazakhstan are its low cost, widespread use in the national economy, and the possibility of complete dissolution of uranium mineralization [4,5]. However, there are negative aspects, such as the high reactivity of the interaction of sulfuric acid with carbonate-, and clay-minerals of ore-bearing rocks. When sulfuric acid interacts with carbonate minerals, gypsum is formed, and clay minerals swell and increase in size; these factors prevent the leaching process [6].

At mining enterprises, the number of production wells and technological blocks increases annually; this is caused by a gradual decrease in the productivity of the blocks being opened and a decrease in the utilization rate of wells, from 9.0 to 7.6. This is due to the accompanying difficulties of opening and mining peripheral parts of blocks with a ragged

ore structure and high heterogeneity of the productive horizon, and low ore filtration coefficients, ranging from 2.0 to 4.0. At the same time, there are no economically justified, effective tools to improve the filtration characteristics of ores and prevent sedimentation for a long period, in difficult mining and geological conditions of well development.

The development of operational blocks in difficult mining and geological conditions is often accompanied by serious complications and an irreversible decrease in the permeability of the downhole zone of the productive reservoir, which dramatically increases the development time and leads to additional costs.

During uranium leaching, the clay content of rocks and ores, as well as the composition of clay cement, largely determines the primary permeability of rocks and reagent consumption. Due to clay minerals, sulfuric acid is neutralized, as a result of which the pH values of solutions do not reach the required parameters (<2.5). This causes an increase in the consumption of chemical reagents, energy, labor, and other operating costs [7,8]. Overcoming the solubility threshold and achieving the required uranium content in PS (Productive Solution) requires an increase in the time period of active leaching, which leads to a decrease in the technical and economic indicators of the process [9].

When designing the borehole development of uranium deposits and feasibility studies, studies of the mineral composition of ores and host rocks of the productive horizon were provided, as well as conducting experimental work on leaching uranium using sulfuric acid solutions with different acidities. Laboratory experiments on uranium leaching were carried out, to determine the effective parameters of the oxidizer application during the intensification of borehole extraction of uranium from ores [10,11]. The experiments included the establishment of mineral and granulometric characteristics of samples, and the leaching of uranium from the core in tubes using solutions with high and standard acidity. A solution with low acidity was prepared separately, but with oxygen feeding into the solution, for conversion from iron (II) to iron (III), for the purpose of effecting subsequent oxidation of uranium (IV) to uranium (VI).

## 2. Materials and Research Methods

### 2.1. X-ray Phase and Granulometric Studies

The research was carried out on the material of core samples from the uranium deposit of the Shu-Sarysu province. Previously, using an atomic emission spectroscope, with individual bound plasma, on the iCAP 7400 spectrometer, all private samples had been analyzed for the content of uranium, aluminum, carbonate, calcium, iron, and magnesium in the rock. The results of the analysis of private samples are shown in Table 1.

**Table 1.** Results of analysis of private samples [10].

| Private Sample | U | $CO_2$ | Al | Ca | Fe Total | Mg |
|:---:|:---:|:---:|:---:|:---:|:---:|:---:|
| 1 | 0.0502 | 0.17 | 4.9640 | 1.2031 | 0.8031 | 0.3694 |
| 2 | 0.0568 | 0.18 | 4.5153 | 1.1953 | 0.8249 | 0.4780 |
| 3 | 0.0296 | 0.10 | 4.6824 | 0.8658 | 1.0250 | 0.3854 |
| Average value, % | 0.0455 | 0.15 | 4.7205 | 1.0881 | 0.8843 | 0.4109 |

The analysis results shown in Table 1 indicate that the $CO_2$ content averages 0.15% of the total sample weight, which indicates a low content of carbonate minerals in the ore and host rocks. However, the average Al content in process samples is 4.7%, which indicates the presence of a significant amount of feldspar and clay minerals. Table 2 shows data on the granulometric composition of the process sample.

Data from the Table 2 shows that the process sample consists mainly of a fine-grained fraction <0.35 mm—57.33%. This indicates an increased content of clay-silt particles, which prevents the filtration of solutions and the process of sulfuric acid leaching.

**Table 2.** Granulometric composition of the process sample.

| Content, %, of Fractions with Fineness, mm | | | | | | | | Total Indicator, % |
|---|---|---|---|---|---|---|---|---|
| Large | | | | Medium | | Small | | |
| >2> | >1.6> | >1.4> | >1> | >0.8> | >0.5> | >0.35> | <0.35 | |
| 1.05 | 2.24 | 1.38 | 3.15 | 1.93 | 6.35 | 15.46 | 57.33 | 100 |

The mineralogical characteristics of the samples were studied using a DRON-3 X-ray diffractometer. The result of the sample diffractometry is shown in Figure 1, the results of the analysis of quantitative and qualitative characteristics are shown in Table 3.

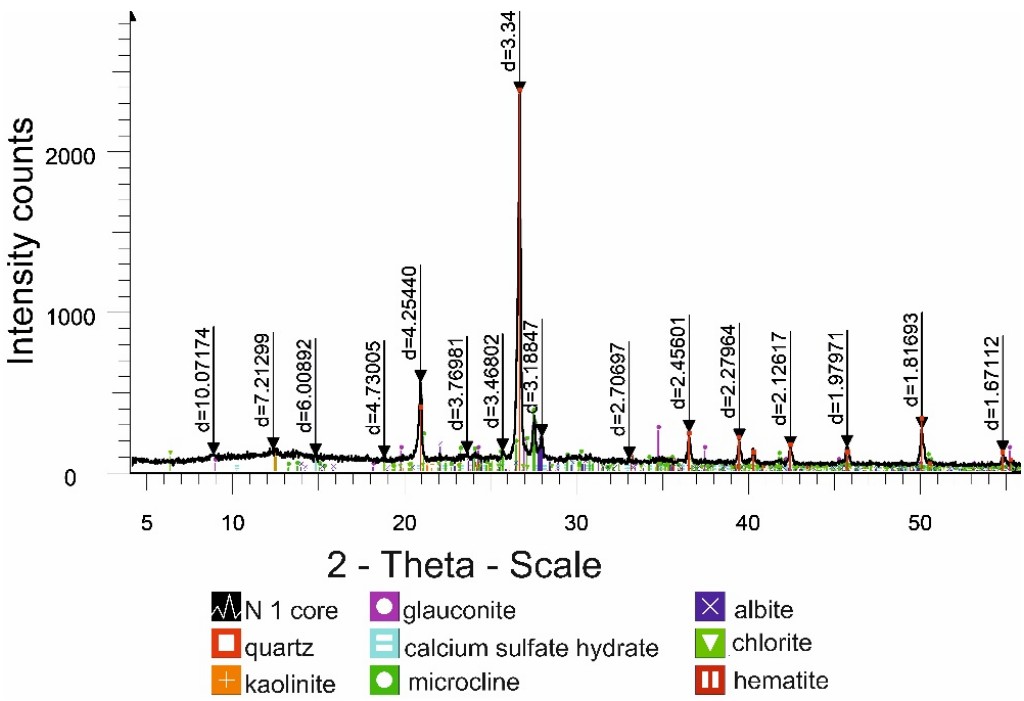

**Figure 1.** Diffractogram of the initial core material sample.

**Table 3.** Results of analysis of quantitative and qualitative characteristics of samples.

| Mineral | Formula | Content, % |
|---|---|---|
| Quartz | $SiO_2$ | 58.6 |
| Kaolinite | $Al_2 (Si_2O_5) (OH)_4$ | 10.3 |
| Glauconite | $(K, Ca, Na)_{0.8}(Al, Fe, Mg)_2(Si, Al)4O_{10}(OH)_2 \, nH_2O$ | 7.9 |
| Calcium Sulfate | $Ca (SO_4) (H_2O)_{0.5}$ | 7.5 |
| Microcline | $(K, Na) AlSi_3O_8$ | 6.4 |
| Albite | $(Na, Ca) (Al, Si)_4O_8$ | 4.1 |
| Chlorite | $(Mg, Fe)_5Al(Si_3Al)O_{10}(OH)_8$ | 2.8 |
| Hematite | $Fe_2O_3$ | 2.4 |

According to the results of the analysis of the quantitative and qualitative characteristics of the samples, given in Table 3, it can be seen that the average quartz content is (58.6%). However, the discovered clay minerals: glauconite (7.9%), kaolinite (10.3%), microcline (6.4%) and albite (4.1%) cause difficulties in the leaching process, due to their swelling during reservoir water displacement and pH changes. The presence of gypsum (7.5%) also forms an obstacle, in the form of precipitation in the places of unloading, namely in the

filter zones of wells, when the speed of movement of solutions changes. The predominance of fine fractions and fine-grained sands in the ores of the productive horizon causes widespread clogging of the pore space and difficulties in restoring the initial filtration of solutions. This explains the difficulties that arise in extracting uranium under similar conditions, with the current range of technological standards: LS (Leaching Solution) injection–PS pumping, and a decrease in the well utilization rate due to downtime for repair and restoration work (RRW) [12]. The X-ray diffractometer is not able to determine the content of uranium minerals, due to the small ratio (less than 1.0%).

### 2.2. Laboratory Studies on Leaching in Tubes from Core Samples

Tube leaching of uranium from samples in laboratory conditions enables the acquisition of more complete information about the process of dissolution of uranium minerals. The flow of the prepared solvent was carried out through a sample of ore material, through filled sealed tubes. The diagram of the laboratory installation is shown in Figure 2. To determine the mass of the sample in the corresponding tube, the technological sample was mixed and weighed on an electronic scale before loading. Filtration of solutions was carried out, with a constant injection of solvent and a pressure drop at the inlet and outlet. To determine the consumption of chemical reagents, a standard for the preparation of solutions with different acidities was developed. In order to study the flow rate and volume of dissolved uranium, relative to the initial sample, the filtered solution was collected in a container, for further analysis.

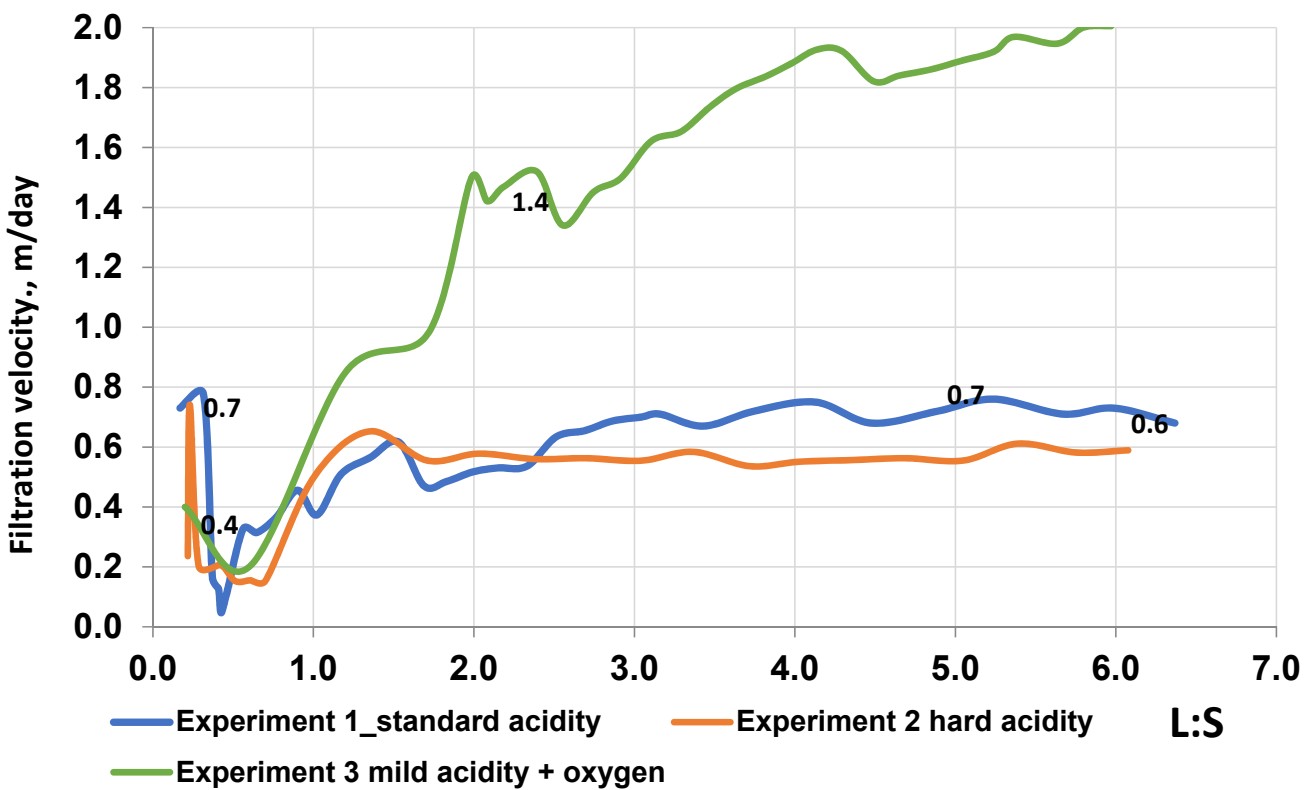

**Figure 2.** Dynamics of the change in the filtration rate of the solution depending on the L:S.

Before applying the solvent, reservoir water was injected into the tubes with a volume of one pore space. For Experiment 1 (tube 1), a standard was adopted with standard acidity in a leaching solution, with a change in the concentration of sulfuric acid in the LS, with a regime: 25–15–10–5–3 g/L when changing L:S from 0 to 2.5. For experiment 2 (tube 2), a standard with a strict acidity regime in the leaching solution was selected, providing for a change in the concentration of sulfuric acid with a regime: 25–20–15–10–5 g/L, with an increase in L:S from 0 to 2.5. In experiment 3 (tube 3), a standard with a mild acidity

regime with the addition of oxygen to the solution under pressure from an oxygen cylinder, provided for the supply of sulfuric acid with a regime: 15–10–5–3 g/L, with a change in L:S from 0 to 2.5. Oxygen was injected into a tank containing leaching solution, as an oxidizer of iron (II), to effect the subsequent oxidation of uranium (IV). Table 4 shows the parameters of the acidity of the solutions in the experiments, according to the developed standards. The L:S indicator is the ratio of a unit volume of liquid leaching solution to a unit volume of solid (ore), it is one of the main indicators for calculation in geotechnological processes. The total time spent on conducting experiments depends on the rate of liquid filtration through the sample and the rate of dissolution of uranium minerals. The time spent on core samples from different horizons or deposits will be different. The rate of passage of solutions through the core material sample depends on the filtration characteristics of rocks, composition of the solution, and sedimentation capacity of the ores [13]. The leaching solution was prepared under laboratory conditions from reservoir water (the uranium deposit in question) and dosed with the addition of sulfuric acid, according to the standards.

**Table 4.** Parameters of acidity of solutions.

| Ratio L:S | Acidity in LS, g/L, According to Experiments | | |
|:---:|:---:|:---:|:---:|
| | 1 | 2 | 3 |
| 0–0.2 | 25 | 25 | 15 + oxygen |
| 0.2–0.8 | 15 | 20 | 10 + oxygen |
| 0.8–1.5 | 10 | 15 | 10 + oxygen |
| 1.5–2 | 5 | 10 | 5 + oxygen |
| 2–2.5 | 3 | 5 | 3 + oxygen |
| 2.5–3 | 0 | 0 | 0 |

According to the developed method of experimental studies, solutions for experiment 1, with standard acidity, provided for the preparation and supply of a solution with the values: 25–15–10–5–3 g/L in the L:S ranges: 0–0.2; 0.2–0.8; 0.8–1.5; 1.5–2; 2–2.5; when L:S > 2.5; with zero acidity. Experiment 2 provided for the preparation and supply of working solutions with a strict acidity regime, with a concentration of sulfuric acid: 25–20–15–10–5 g/L at the appropriate L:S ranges. In experiment 3, working solutions were prepared and fed under a mild acidity regime, with the values: 15–10–10–5–3 g/L when compressed oxygen was added in the L:S ranges: 0–0.2; 0.2–0.8; 0.8–1.5; 1.5–2; 2–2.5; when L:S > 2.5; without the addition of acid.

The parameters of the oxidizer's effect on the process of uranium dissolution from samples were determined using output solutions, which were recorded in a log, with volume determination in measuring vessels, for further calculations and analyses. Using the obtained data from the output solution, the volume of dissolved uranium was calculated, in addition to the values of L:S, pH, ORP, residual acidity, $Fe^{3+}$ content, $Fe^{2+}$ content and specific consumption of sulfuric acid per 1 kg of uranium. The content of uranium and iron in the solution was clarified and checked by titration, according to the method of chemical measurements (MVI) # 36-2019 # KZ06.01.00050-2019 from 11.07.2019.

## 3. Discussion or Results

The process of leaching uranium in natural permeability is a complex process involving several stages, one of which is heterogeneous, controlling at the liquid-solid phase boundary, containing a reagent capable of forming highly soluble compounds when interacting with uranium-containing minerals or rocks [14]. The calculated data obtained, based on the results of laboratory experiments, made it possible to plot: the change in the solution flow rate, as a function of L:S (Figure 2); the values of the uranium concentration in productive solutions, as a function of L:S (Figure 3); andthe extraction of uranium, as a function of

L:S (Figure 4); with the flow rate of the chemical reagent depending on L:S (Figure 5). The flow rate of the solution in the ore during the processes of interaction with uranium minerals, with further dissolution and transfer to the unloading and lifting zone, is one of the important parameters and is determined by the ore filtration coefficient [15,16]. The $K_f$ ore filtration coefficient is determined in (m/day) and is calculated using the formula:

$$K_\Phi = \frac{\Delta V L}{\Delta t \Delta H S} \quad (1)$$

where: $\Delta V$ is the volume of filtered solvent; $L$ is the length of the tube; $\Delta t$–is the sample measurement time; $\Delta H$ is the hydrostatic pressure drop; $S$ is the cross-sectional area of the tube.

As can be seen from the graph, the changes in the filtration coefficient of $K_f$ depend on the L:S. In experiment 1, when feeding working solutions with a standard mode of acidity, at first the $K_f$ is 0.73 and rises to 0.78, but then it decreases to a minimum value of 0.11 at values of L:S, in the range from 0 to 0.41, with the corresponding acidity parameters of 25–15 g/L. This decrease indicates the formation of colmatation effects, during the interaction of sulfuric acid solutions with ore-containing rocks, that prevent the filtration of solutions. The subsequent gradual increase in the filtration coefficient on the tube from 0.45 to 0.62 at the L:S range, in the range of 0.4–1.4, is due to a decrease in acidity in the working solutions, from 15 to 10 g/L, and a partial reduction in the effect of colmatation. A further decrease in acidity in the solutions did not affect the filtration volume of the solution.

The filtration parameters in experiment 2, with a strict acidity regime of working solutions, approximately correspond to the values of the filtration coefficient in experiment 1, the only dissimilarity being due to a slight difference in acidity (5 g/L). However, the low acidity value in solutions in experiment 3 with the addition of oxygen as an oxidizer positively affected the filtration parameters of the experiment. In experiment 3, with an initial $K_f$ of 0.42 and a further decrease to 0.25 with a change in L:S in the range of 0–0.5, acidity is observed in the working solution of 15 g/L, which indicates a slight formation of the colmatation. A subsequent increase of $K_f$ from 0.25 to 0.9 at values of L:S from 0.5 to 1.5, with an appropriate acidity of 10 g/L, confirms a decrease in the effect of colmatation in the filtrate. The increased values of $K_f$ in experiment 3, compared to experiments 1 and 2, indicate a proportional effect of acidity on the filtration rate and the formation of colmatation effects. The total time spent on conducting experiments, depending on the filtration rate of solutions in core samples, ranged from 320 to 428 h: a total of 320 h were spent on conducting experiment 1; 412 and 428 h were spent on experiments 2 and 3, respectively. A comparative analysis of the results of uranium leaching from core samples using various parameters of solutions allows us to determine the effectiveness of leaching.

These changes in the concentration of the useful component in the solution depending on the L:S, shows the maximum and minimum concentrations of uranium in the productive solution, with the corresponding volume of the output solution. The concentration of uranium in productive solutions of the $C_{cp}^U$ and the ratio of L:S are determined by the formulas:

$$C_{cp}^U = \frac{\sum_{i=1}^n C_i^U \Delta V_i}{\sum_{i=1}^n \Delta V_i} \quad (2)$$

where $n$ is the amount of sampling for a given value; $\Delta V$ is the amount of productive solution in the $i$-th sample; $C_i^U$—the concentration of uranium in the $i$-th specific sample. Counting is performed for all registered samples of productive solution $n$.

As can be seen from the graph (see Figure 3), in experiment 1, when leaching using solutions with a standard acidity regime, the maximum uranium content of 520 mg/L is achieved at L:S 0.561, after overcoming the threshold of 250 mg/L and reducing to 125 mg/L. This indicates the gradual overcoming of the solubility threshold and the achievement of active leaching with the required volume of solution and acid. However, there is a decrease in the concentration of uranium in the productive solution, from 520 mg/L to a low value of 30 mg/L, at a range of values of L:S from 0.7 to 1.5. This may be due to

a decrease in the acidity of working solutions in 10 g/L. In experiment 2, in the process of leaching under a strict acidity regime, the uranium content in the productive solutions reaches a peak of 670 mg/L within the range of L:S 0.7, with a further sharp decrease in the uranium content in the productive solutions from 670 to 130 mg/L, with an increase in L:S to 1.0, which indicates a decrease in acidity from 25 to 15 g/L. In experiment 3, when leaching uranium with a mild acidity regime of solutions and the addition of oxygen, the uranium content gradually increases to 220 mg/L with a corresponding increase in L:S from 0 to 2.0. This also confirms the gradual overcoming of the solubility threshold by weak sulfuric acid solutions, followed by oxidation and conversion of tetravalent uranium into a soluble form, while creating favorable conditions for stable active leaching and oxidation. The uranium content in the solutions in all experiments showed similar maximum peaks, approximately corresponding to the initial uranium content, depending on the acidity of the leaching solutions. This indicates the neutralization of sulfuric acid during interaction with ore-containing rocks and the influence of an oxidizer during the transfer of uranium mineralization into solution (experiment 3).

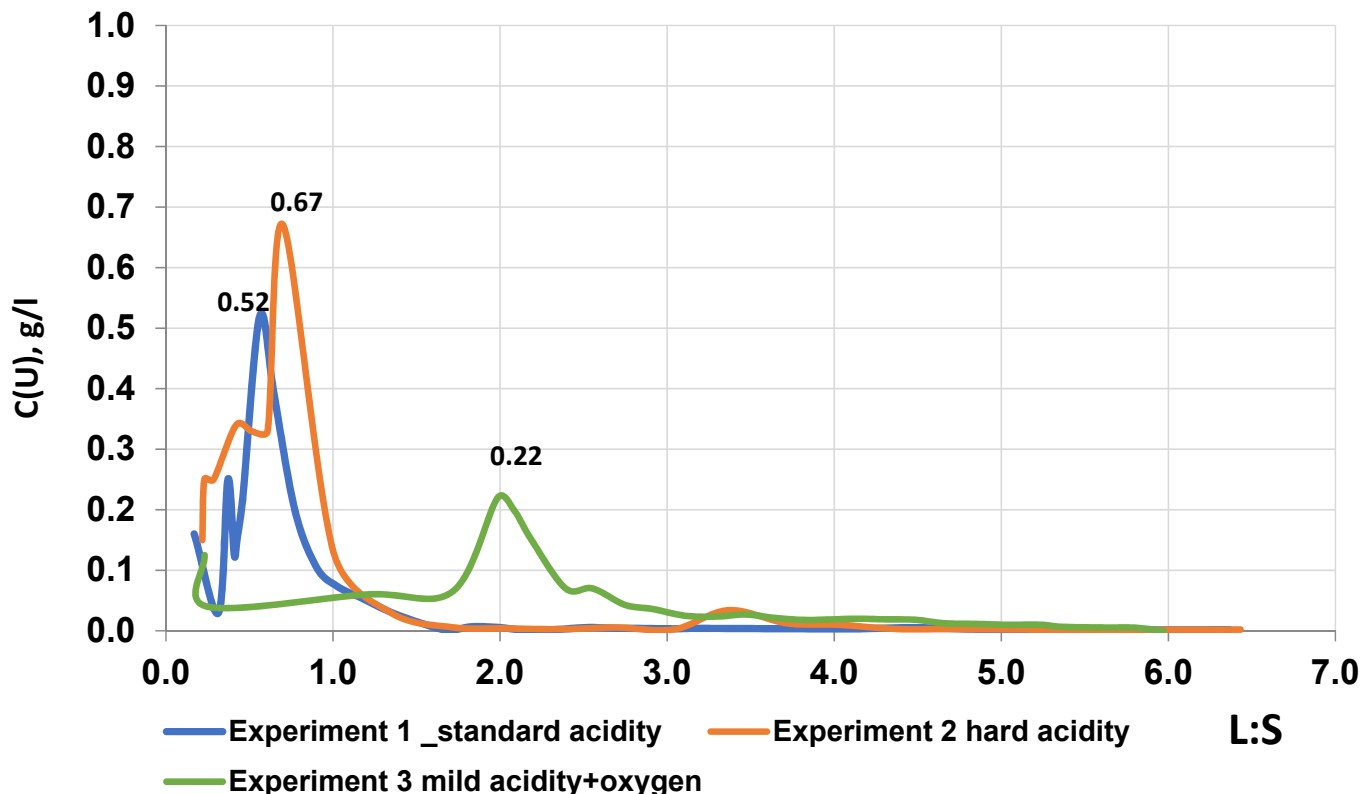

**Figure 3.** Uranium concentration values in productive solutions as a function of L:S.

The effectiveness of the parameter of the proposed solution, with reduced acidity and with the addition of an oxidizer, was determined on the basis of a comparative analysis with the parameters of uranium extraction under standard and rigid modes of acidity of solutions. For this purpose, experimental data were collected and processed, calculations were made and schedules for uranium extraction were constructed. Uranium extraction depends on the uranium content in the productive solution and the volume of the most productive uranium; these data are the most informative and fundamental in assessing the solubility and stability of uranium minerals. Uranium extraction is calculated as the ratio of the mass of uranium in the output solutions to its initial calculated mass in the sample:

$$\varepsilon = \frac{\sum_{i=1}^{n} C_i^U \Delta V_i}{M_p C_{initial\ core\ sample}^U} \tag{3}$$

where $C_{initial\ core\ sample}^{U}$—the concentration of uranium in the initial sample; $C_i^U$—is the concentration of uranium in the productive solution of the $i$-th sample; $M_p$ is the mass of ore in the initial sample; $\Delta V_i$ is the volume of the productive solution in the $i$-th sample.

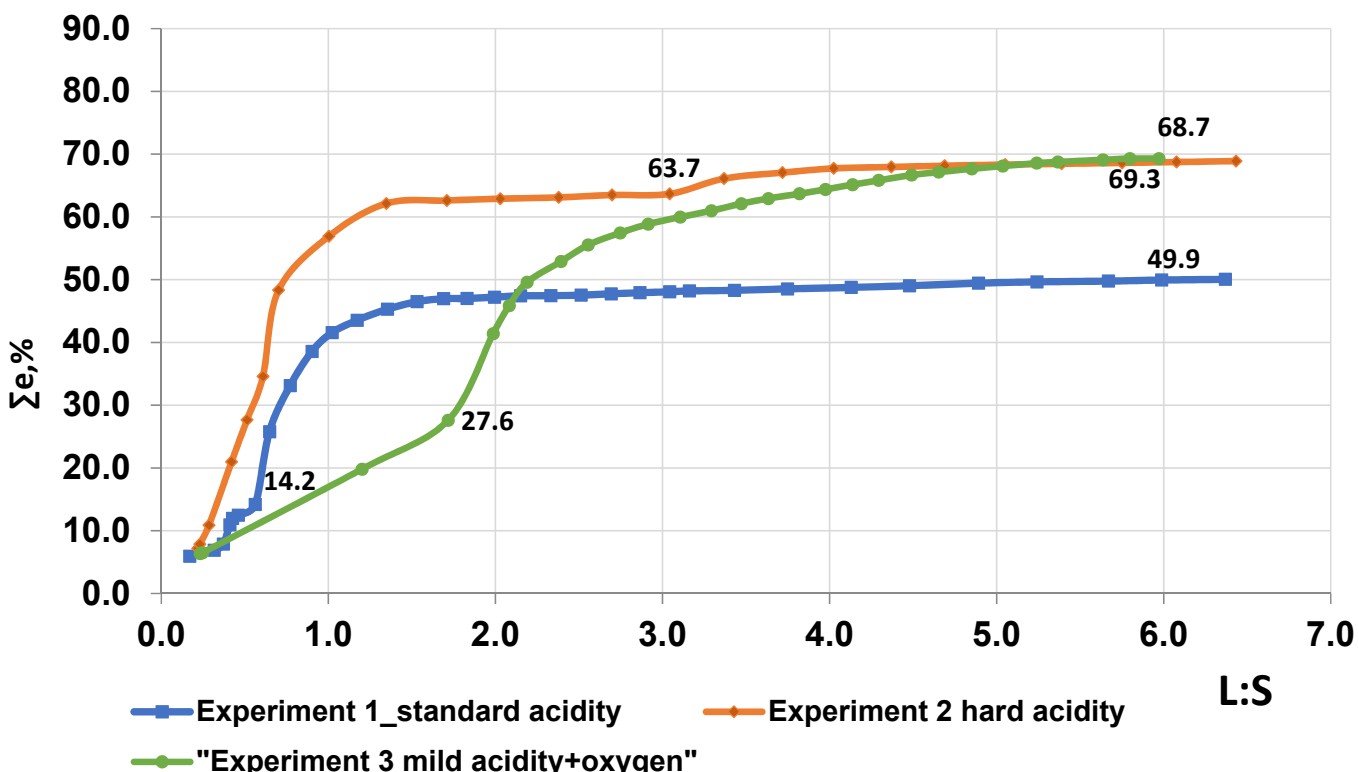

**Figure 4.** Uranium extraction depending on L:S.

The graph in Figure 4 shows that the maximum extraction values in Experiment 1 reached 49% of the mass of uranium contained in the sample; this indicates insufficient solvent capacity of working solutions with standard acidity and low filtration values. A sharp increase in extraction occurs in the range of L:S from 0.5 to 0.9 to 38.6%, followed by a slowdown in extraction at L:S > 1; the acidity of working solutions in this range was maximum: 25–15 g/L. The extraction data correlate with the graph of the uranium content in the solution, where the peak of the maximum uranium content in the PS falls in the range of L:S from 0.5 to 0.8. A further slowdown in the extraction of the useful component is due to a decrease in the uranium content in the solution, due to a decrease in the acidity of working solutions from 15 to 10 g/L. In experiment 2, the increase in uranium extraction occurs at L:S parameters from 0 to 1.0 with the achievement of 56% extraction, followed by a slowdown of 63%, which indicates intensive extraction of uranium in solutions with a strict acidity regime. In the range of L:S from 0 to 1.0, the acidity of the solutions supplied to tube 2 was maximum and amounted to: 25–20–15 g/L. Intensive extraction is due to the high uranium content in the solution: 320–670 mg/L and the corresponding filtration coefficient of the solution is 0.34–0.44 in the range of L:S from 0 to 0.9. The extraction of the useful component in experiment 3, when feeding working solutions with a mild acidity regime with the addition of oxygen, reached a maximum value of 68% with a change in L:S to 3.0. However, the active extraction of uranium in Experiment 3 also occurred in the range of L:S from 1.7 to 3.0, from 27.6 to 60.3%, respectively. The intensive extraction of uranium in experiment 3 in the range L:S 1.7–3.0 is due to the achievement of the solubility threshold and the creation of conditions for active leaching, with the oxidation of iron (II) and the interaction of iron (III) for the oxidation of uranium (IV) [17,18]:

$$4Fe^{2+} + O_2 + 4H^+ \rightarrow 4Fe^{3+} + H_2O \tag{4}$$

$$UO_2 + 2Fe^{3+} + 3SO_4^{2-} \rightarrow UO_2(SO_4) + 2Fe^{2+} + 2SO_4^{2-} \tag{5}$$

To determine the economic efficiency of using an oxidizer in the intensification of uranium leaching, and a comparative analysis of solvent consumption, graphs were constructed based on the experiments performed. The consumption of sulfuric acid as a solvent per kilogram of extracted uranium $P_K$ is calculated as the ratio of the total mass of the reagent consumed during the experiment to the calculated mass of extracted uranium:

$$P_K = \frac{\sum_{i=1}^{n}\left(C_o^K - C_i^K\right)\Delta V_i}{\sum_{i=1}^{n} C_i^U \Delta V_i} \tag{6}$$

where $C_o^K$ is the initial concentration of the reagent in the leaching solution (determined when using a mother liquor with the initial acidity); $C_i^K$—the actual concentration of the reagent in the $i$-th sample of the leaching solution (actual); $C_i^U$ the actual concentration of uranium in the productive solution of the $i$-th sample; $M_p$ the total mass of ore used in the experiment; $\Delta V_i$ the total volume of leaching solution in the $i$-th sample.

Reagent costs per unit of processed ore mass (calculated to determine the acid capacity of ore) are determined by the formula:

$$P_K = \frac{\sum_{i=1}^{n}\left(C_o^K - C_i^K\right)\Delta V_i}{M_P} \tag{7}$$

where $C_o^K$ is the initial concentration of the solvent reagent in the leaching solution (determined when using sorption curves with residual acidity); $C_i^K$—is the concentration of the solvent reagent in the $i$-th sample of the leaching solution (actual); $M_p$ is the mass of ore used in the experiment; $\Delta V_i$ is the volume of the leaching solution in the $i$-th sample.

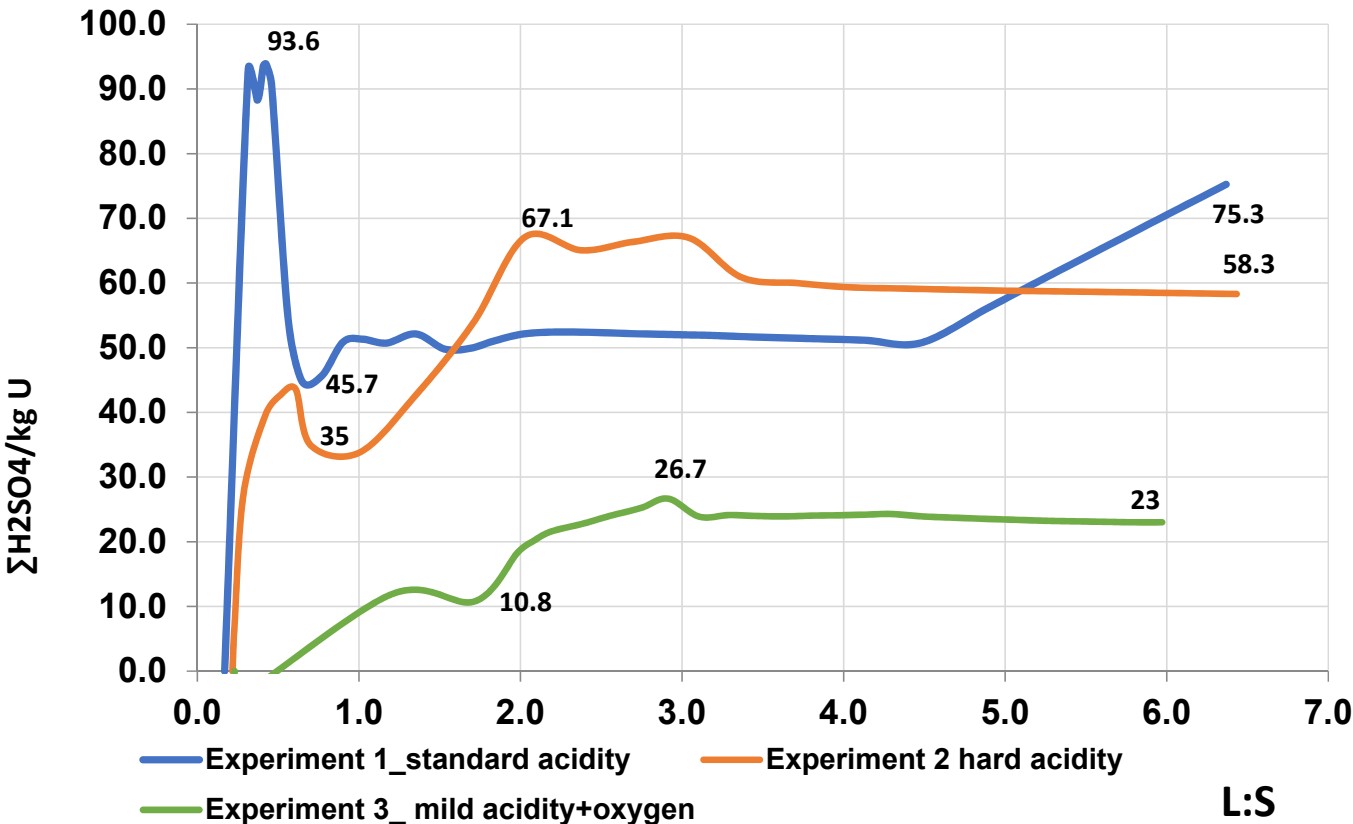

**Figure 5.** Consumption of sulfuric acid depending on L:S of the process.

The graph in Figure 5 shows that the consumption of sulfuric acid in Experiment 1 reaches maximum values of 90–93 kg/kg U in the L:S range: 0.2–0.7, after which it sharply decreases to values of 48–50 kg/kg U. A sharp increase in the consumption of the solvent reagent is associated with the acidity in the leaching solution and low parameters of uranium extraction in the range of L:S from 0 to 0.8. After increasing the extraction of uranium from 25.8 to 47% in the L:S range from 0.647 to 1.689, there is a noticeable decrease in the consumption of the solvent reagent. The maximum values of the reagent consumption in experiment 2, when feeding working solutions with a hard acidity regime, reach 67 kg/kg U at L:S 2.0. A gradual increase in the specific consumption in experiment 2, with a hard acidity regime of working solutions, is due to a slowdown in the extraction of uranium in the corresponding range L:S. The maximum values of the specific consumption of sulfuric acid in experiment 3 are not able to reach the corresponding indicators in experiment 2 and amount to 26.7 kg/kg U with a change in L:S from 2.0 to 3.0, after which they decrease to the minimum values of 23 kg/kg U. Reduction of reagent consumption in the L:S range >3.0 is due to intensive uranium extraction in this interval. In a comparative analysis of the solvent reagent consumption, the lowest values are observed in Experiment 3, with a mild acidity regime of working solutions and with the addition of oxygen as an oxidizer. The decrease in reagent consumption is caused by intensive uranium extraction, due to the oxidizing effect of iron (III) on uranium (IV) in its conversion to uranium (VI), during its further dissolution. In the hard mode of acidity (experiment 2), the specific acid consumption is higher than in the soft mode of acidity, with the addition of an oxidizer, but lower than in the standard mode of acidity (experiment 1), due to the intensive extraction of uranium from the tube.

## 4. Conclusions

X-ray phase studies of ore-bearing rocks indicate their diverse mineral composition, and the presence of calcium sulfate will contribute to chemical colmatation in wells. The predominance of a fine fraction of fine-grained sands (57% of the total mass) prevents the filtration of solutions in the productive horizon. The data indicate a complex structure of the productive horizon with low filtration parameters. As practice shows, in such conditions, the leaching processes are hindered by the low uranium content in the PS and low parameters of the turnover of solutions, which requires continuous intensification of the process by hydrodynamic methods.

The data indicate the direct influence of the concentration of sulfuric acid on the dissolution of uranium minerals and the content of uranium in productive solutions. However, the influence of filtration characteristics on the overall indicators of the process (extraction, specific acid consumption) is no less important when solving issues of production intensification. The addition of oxygen as an oxidizer to sulfuric acid solutions, with reduced acidity, positively affects the intensification of uranium leaching. Reducing the acidity of working solutions significantly increases the filtration characteristics of samples, due to the oxidizing ability of iron (III) to uranium (IV); the uranium content in the solutions reach the design values of 300 mg/L. A decrease in the consumption of the solvent reagent (37 kg $^{H2SO4}$/kg $^{U}$) indicates the economic feasibility of using an oxidizer at the first stage of well production, in areas with a predominance of ores containing uranium (IV). In addition, the low acidity of the leaching solution will not create sedimentation, will lead to intensive development of blocks and reduce operating costs, electricity costs and labor costs.

These studies allowed us to establish that the uranium content in the productive solution directly depends on the concentration of acid in the leaching solution. However, an increase in the concentration of the solvent reagent does not always lead to an increase in its consumption. The analysis of the values of uranium extraction against time allows us to determine the optimal concentration and consumption of reagents during borehole production. The optimal concentration is one in which an excess leads to a sharp consumption of acid and a slight increase in the intensity of uranium extraction. The value of the

optimal acid concentration depends on the properties of the specific ore minerals in the test samples.

The use of air at the enterprises' uranium mining, in areas with low filtration characteristics of ores, has become possible thanks to special devices, the design of which allows for dosing of air into the LS pipeline. This technology does not need capital investments and allows for reductions in the consumption of chemical reagents, as well as increases in the extraction of uranium, by improving the filtration characteristics of ores and host rocks. Scientists and specialists at Satbayev University have already developed this area and have achieved positive results [19].

**Author Contributions:** Conceptualization, B.R. and Z.K.; methodology, Z.K. and M.M.; software, M.M. and K.T.; validation, Z.K., M.M. and K.T.; formal analysis, Z.K. and M.M.; investigation, Z.K. and K.T.; resources, Z.K. and K.T.; data curation, B.R.; writing—original draft preparation, Z.K. and K.T.; writing—review and editing, Z.K. and K.T.; visualization, Z.K. and K.T.; supervision, B.R.; project administration, B.R.; funding acquisition, B.R. All authors have read and agreed to the published version of the manuscript.

**Funding:** This research is funded by the Science Committee of the Ministry of Education and Science of the Republic of Kazakhstan (Grant No. AP08856422—"Development of an Innovative Technology for the Intensification of Downhole Uranium Mining Using Hydrodynamic Decalmatation Device in Combination with a Complex of Chemicals for Multifunctional Purposes").

**Data Availability Statement:** Not applicable.

**Acknowledgments:** The authors express their gratitude to the Limited Liability Partnership "Institute of High Technologies" at JSC "NAC "Kazatomprom" for providing the computer program and for the possibility of using the laboratory facilities.

**Conflicts of Interest:** The authors declare no conflict of interest. The funders had no role in the design of the study; in the collection, analyses, or interpretation of data; in the writing of the manuscript, or in the decision to publish the results.

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
