# Peer review of "Improving the Efficiency of Downhole Uranium Production Using Oxygen as an Oxidizer"

_minerals, doi:10.3390/min12081005_

Round 1

Reviewer 1 Report

The features occurring during borehole uranium mining in deposits with low filtration characteristics, as well as the conditions and reasons for the reduction of geotechnological parameters of uranium mining by the well are considered. Core material samples were taken from the productive horizon of the Chu-Sarysui province deposit and granulometric compositions were established. The contents of uranium, aluminum, calcium, magnesium, iron and carbonate minerals in the samples were determined by atomic emission spectroscopy. The X-ray phase analysis method established the features and quantitative and qualitative characteristics of ore-containing minerals. 

Comments:

Page 1: Line 17: "conducting experimental experiments" this is redundant....changes

Page 2: Table 1 and line 87: CO2 is not carbonate

Page 3: Table 2,  the way to express particle sizes is not clear

Figure 1: There are names of minerals that are not in English

Table 3: The samples do not have uranium??

Correct contents of glauconite, calcium sulfate, microcline and albite

Page 4: Table 4, from the ratio L.S column...remove "of"

Page 11: 37 kg/kg ...what do you mean?..should be clear

Question: What is the leaching time??...it doens't matter??

Author Response

We corrected everything according to your comments

Reviewer 2 Report

Comments for minerals-1812216-R1

Although the author made some revisions, I did not see any traces of modification. The entire study is incomplete and not suitable for publication in its current state. The main problems are as follows: 

1. Lack of leaching time, leaching temperature and other technological parameters on the effect of leaching. 

2. Lack of characterization of leaching residue, such as: main chemical composition, scanning electron microscopy, morphology and other basic tests. 

3. Lack of comprehensive condition test results. 

4. Other details also need to be modified, especially the specification of graphs and tables. For example, the chemical content "%" in Table 1 is unnecessary to add "%" after each ingredient.。

Author Response

(The authors gave the same response as above.)

Reviewer 3 Report

The paper describes an experiment taken place in a tubular reactor simulating the leaching behavior of U in a solution mining situation. It reads well. The legend of Fig 1 is incomplete.

Author Response

(The authors gave the same response as above.)

Round 2

Reviewer 1 Report

Comments:

In leaching, time is important, therefore, it is required to know what the leaching time is.

Author Response

We corrected everything

Reviewer 2 Report

Comments for minerals-1812216

Although the author has made some revisions and adjustments, as an academic paper, the research content needs to be basic.  The author still does not provide some key experimental content, and the main problems are as follows: 

1. Supplement the content information of main chemical components of leach residue after leaching. 

2. Please add XRD analysis pattern of leached residue and make necessary analysis.  It would be better to supplement the XRD analysis of each test condition, which would be more beneficial to the reader. 

3. The author needs to give some explanation on whether the author will apply oxygen in the future, how to add oxygen, or replace oxygen with air.

Author Response

We corrected everything

This manuscript is a resubmission of an earlier submission. The following is a list of the peer review reports and author responses from that submission.

Round 1

Reviewer 1 Report

Comments:

- Tables 3 and 4 are not in the format of the Minerals

- Remove the commas from equations 1, 5 and 6

- What criteria were used to establish the L:S ratios?

- The work is simple and the number of references must be increased

Reviewer 2 Report

Comments for minerals-1734235

The topic of this paper is meaningful and has certain practical application value.  The author also carried out a large number of experimental studies, the overall situation is good.  However, there are also many problems. Some experimental data are missing. The main problems are as follows:

1. The abstract is not well expressed, and key technical parameters and main quantitative results need to be added.

2. The content of Introduction is too small, which requires the author to supplement the research related to the research topic, especially relevant specific data.

3. There are problems with the data in Table 3, such as 7.9.9, 7.5.5, 6.4.4 and 4.1.1, which are clearly wrong and need to be modified.  In addition, the table is missing a total row for all minerals, and the total mineral content should be 100% or close to 100%. "Concentration" is not appropriate, but "content" is more reasonable.

4. The meaning of "Ratio L:S" in Table 4 should be stated when it first appears in the article.

5. Discussion of the results. Using “Discussion or results and Discussion is better”.

6. The repeated experiment under comprehensive conditions is lacking, which needs to be supplemented by the author.

7. Lack of main component data of leach solution and infrared spectrum and other partial test characterization of leach solution.

Reviewer 3 Report

The paper describes an experiment taken place in a tubular reactor simulating the leaching behavior of U in a solution mining situation. It is written in a confusing manner and is very difficult to follow the procedure. The reproducibility of the experiments is questionable. The authors should know there are tons of references in chemical reactions taken place in packed beds and use the models used there.

The authors should learn how to write the abstract. The Abstract should be clear and concise, typically 200-250 words maximum. It should include the purpose, a brief description of the work, and the pertinent results or conclusions.

Table 1: The authors indicated that the results in Table 1 were performed previously…When and where – give a reference. Use period instead of comma. Use three significant figures. The authors have used the comma and period interchangeably throughout the paper and should be corrected.

Table 3: Correct 7.9.9; 7.5.5; 6,4,4; 4.1.1

Line 105: The diagram of the laboratory installation is shown in Figure 2. – Misleading and untrue.

The experimental procedure given in line 102-120 is confusing and should be rewritten. What do you mean by the standard, hard acid solutions? Did the strength of the acid change with the LS ratio and how did the change affect the dissolution of U? How does the sample measurement time defined? Is it different from the residence time? What were the pHs of the solution before and after the reaction?

Table 4: It is very confusing. For example, in the 2nd column, 25, 15, 10 etc., is confusing. Does it mean that 25 g of concentrated sulfuric acid in one liter of water? What are the initial and final pH’s of these solutions? What does it mean by 15+oxygen? What oxygen and how much? What are the porosity of the tubular reactor? How did the intra and inter porosity change during and after the leaching studies?

Fig 2: Is filtration velocity the same as the filtration rate? If it is, the units are inconsistent. The meaning of Eq 1 is unclear.

Line 163: Define the filtration coefficient.

How and where Eqs 2 and 3 were used?

Complete Eqs 5 and 6.

Eq 8 is unnecessary! Just say that the denominator of Eq 7 is assumed to be Mp.